# Development and Validation of a UPLC-MS/MS Method for the Quantification of Components in the Ancient Classical Chinese Medicine Formula of Guyinjian

**DOI:** 10.3390/molecules27238611

**Published:** 2022-12-06

**Authors:** Nan Ge, Zhineng Li, Le Yang, Guangli Yan, Aihua Zhang, Xiwu Zhang, Xiuhong Wu, Hui Sun, Dan Li, Xijun Wang

**Affiliations:** 1National Chinmedomics Research Center, Heilongjiang University of Chinese Medicine, Heping Road 24, Harbin 150040, China; 2State Key Laboratory of Dampness Syndrome, The Second Affiliated Hospital Guangzhou University of Chinese Medicine, Dade Road 111, Guangzhou 510120, China; 3Beijing-Tianjin-Hebei Lianchuang Drug Research (Beijing) Co., Ltd., No. 100, Balizhuang Xili, Chaoyang District, Beijing 100025, China; 4State Key Laboratory of Quality Research in Chinese Medicine, Macau University of Science and Technology, Avenida Wai Long, Macau 999078, China

**Keywords:** Guyinjian, UPLC-MS/MS, quantitative analysis, multivariate statistical analysis, quality control

## Abstract

Guyinjian (GYJ) is an ancient classic formula of traditional Chinese medicine used for the treatment of liver and kidney yin deficiency; it was derived from the book “Jing Yue Quan Shu” in the Ming Dynasty. Modern clinical observation experiments have shown that GYJ has a definite therapeutic effect on the treatment of gynecological diseases such as kidney deficiency type oligomenorrhea, climacteric syndrome, intermenstrual bleeding, pubertal metrorrhagia, etc. However, the lack of GYJ quality control studies has greatly limited the development of its wider clinical application. In this study, a validated UPLC-MS/MS method was developed successfully for the first time and used to quantify fourteen compounds in GYJ samples with good specificity, linearity (r = 0.9960−0.9999), precision (RSD% ≤ 3.18%), stability (RSD% ≤ 2.22%) and accuracy (recovery test within 88.64–107.43%, RSD% at 2.82–6.22%). Simultaneously, the determination results of 15 batches of GYJ samples were analyzed by multivariate statistical methods, and it was found that the compounds have a greater influence on batch-to-batch stability, mainly Rehmannioside D, Loganin, Morroniside, Ginsenoside Re, and 3′,6-Disinapoylsucrose. The proposed new method has the advantages of high sensitivity, high selectivity, and rapid analysis, which provides a reference for the GYJ quality control study.

## 1. Introduction

Guyinjian (GYJ), a classical Chinese medicine formula, is derived from the book “Jing Yue Quan Shu”, Volume 51, written by Zhang Jingyue in the Ming Dynasty (1640 A.D.). The original book stated that “This recipe is composed of Ginseng Radix et Rhizoma, Rehmanniae Radix Praeparata, Dioscoreae Rhizoma, Corni Fructus, Polygalae Radix Praeparata, Glycyrrhizae Radix et Rhizoma Praeparata Cum Melle, Schisandrae Chinensis Fructus, and Cuscutae Semen, and it is effective in tonifying the liver and kidney, nourishing Yin and strengthening essence” [1]. In recent clinical observation experiments, it was found that addition and subtraction therapy of GYJ can increase ovarian blood supply, improve high ovarian reserve function, reduce Gn (gonadotropins) consumption, alleviate symptoms of kidney yin deficiency, and can ameliorate ovarian responsiveness and pregnancy outcome [2]. Furthermore, GYJ can also be used to treat polycystic ovary syndrome ovulation disorders [3], immuno-sterility [4], kidney deficiency type oligomenorrhea [5,6], climacteric syndrome [7], intermenstrual bleeding [8], pubertal metrorrhagia [9] and other gynecological diseases, all with good therapeutic effects. Simultaneously, GYJ has been presented in the Catalogue of Ancient Classical Formula (First Batch) promulgated by the National Administration of Traditional Chinese Medicine, which provides policy support for the further development of granules [10]. Although the clinical applications and pharmacological effects of GYJ have been well explored, the quality control studies are still scant. Therefore, it remains necessary to develop a set of comprehensive and rapid quality evaluation methods for qualitative and quantitative analysis of the major components in GYJ samples.

The index compounds were selected via a literature review, to find the main components of each medicine that exert pharmacological activities; these refer to the quality control components in the “Pharmacopoeia of the People’s Republic of China” (2020 edition). Those ultimately selected to be the index components of the GYJ samples were Ginsenoside Re, Ginsenoside Rg1, Ginsenoside Rb1 from Ginseng Radix et Rhizoma; Rehmannioside D and Verbascoside from Rehmanniae Radix Praeparata; Morroniside and Loganin from Corni Fructus; 3′,6-Disinapoylsucrose, Polygalaxanthone III and Tenuifolin from Polygalae Radix Praeparata; Hyperoside from Cuscutae Semen; Schisandrin from Schisandrae Chinensis Fructus; Liquiritin and Glycyrrhizic acid from Glycyrrhizae Radix et Rhizoma Praeparata Cum Melle. The structures of the fourteen constituents are shown in Figure 1.

The development and application of mass spectrometry provides a quick and convenient method for the identification and quantification of contents in complex natural medicine extracts, with its superior sensitivity and resolution [11]. Triple quadrupole mass spectrometry has been widely used in quantitative studies because of its higher accuracy [12]. The Selected Ion Recording (SIR) mode is a quantitative analysis mode for a selected ion, which allows simultaneous quantitative analysis of multiple components with only one quadrupole [13]. In this study, based on UPLC-MS/MS technology, a rapid, simplified and efficient method for the quantitative analysis of fourteen components of GYJ samples was successfully developed for the first time and methodologically validated. In addition, the determination results of 15 batches of GYJ samples were analyzed by multivariate statistical methods; it was found that the compounds with a greater influence on batch-to-batch stability were mainly Rehmannioside D, Loganin, Morroniside, Ginsenoside Re, and 3′,6-Disinapoylsucrose. This study will contribute to the quality control study for GYJ and its preparations.

## 2. Results and Discussion

### 2.1. Optimisation of Sample Extraction Conditions

To obtain the optimal quantitative extraction, the extraction methods including reflux extraction and ultrasonic extraction were investigated; it was found that there was no significant difference in extraction efficiency between the two methods, so the more convenient ultrasonic extraction method was selected. The preliminary experiment (simultaneous determination of six chemical components (Morroniside, Loganin, Hyperoside, 3′,6-Disinapoylsucrose, Glycyrrhizic acid, and Schisandrin) in GYJ based on HPLC (UV detector)) investigated pure water, 25% methanol, 50% methanol, 75% methanol, 100% methanol, 25% ethanol, 50% ethanol, 75% ethanol, and 100% ethanol, and found that 75% methanol was the best. Therefore, in the process of this experiment, only 50% methanol, 75% methanol, and 100% methanol were investigated when designing the extraction solvent. However, due to the increase of the determining components, the factors to determine the optimal conditions also increased. The results were analyzed comprehensively, and it was discovered that the content of most of the components does not have much variation under the conditions of 50% methanol and 75% methanol; however, Rehmangoside D has better water solubility and is an important index component of the monarch drug Rehmanniae Radix Praeparata. Therefore, the more suitable 50% methanol was selected as the best extraction solvent. The extraction time was investigated for 15 min, 30 min, 45 min, and 60 min, correspondingly, and the results have proven that 45 min of ultrasonic extraction could achieve full extraction. The solid–liquid ratio of 2 mg/mL, 4 mg/mL, 8 mg/mL, and 16 mg/mL was respectivel investigated, and the experimental results show that 8 mg/mL was more suitable. In summary, the final sample extraction conditions were a precise sampling of 0.4 g, precise addition of 50 mL of 50% methanol, and ultrasonic extraction for 45 min.

### 2.2. Optimisation of UPLC-MS/MS Condition

The sample solution was firstly subjected to full scan analysis in MS Scan mode. It was found that among the fourteen interest monitoring components, only Schisandrin did not respond in the negative ion mode, so Schisandrin was selected to be monitored in the positive ion mode, and the remaining thirteen compounds had good response values in the negative ion mode. The SIR parameters of the fourteen compounds were individually optimized to achieve the highest sensitivity and resolution. The optimum cone voltage was determined by comparing the peak areas of each compound at different cone voltages. Taking Schisandrin as an example, the cone voltages were set to 20 v, 25 v, 30 v, 35 v, 40 v, and 45 v, individually. It was found that the peak area of the compound first increased and then decreased with the increase of cone voltage, and when the cone voltage of Schisandrin was 30 v, the peak area was the largest. The summary results of the optimum cone voltages are shown in Table 1. In order to obtain a chromatogram with a good separation effect, the elution gradient of the chromatogram was in the first place. After optimization, Schisandrin appeared in 2.07 min in positive ion mode, and 13 components were successfully separated in 14 min in negative ion mode. Afterwards, the conditions of column temperature (30 °C, 35 °C, 40 °C), flow rate (0.3 mL/min, 0.4 mL/min), injection volume (5 µL, 2 µL, 1 µL) were investigated. It was found that the best separation results were obtained when the column temperature was 35 °C, the flow rate was 0.4 mL/min, and the injection volume was 1 µL. Representative chromatograms are shown in Figure 2.

### 2.3. Validation of UPLC-MS Method

The results of the specificity experiment showed that the method used for the determination of the fourteen components had no interference from other compounds, and the specificity was excellent. All chromatogram comparison results are shown in Appendix A. All calibration curves were constructed by plotting the peak area (y) versus the concentration (x: in µg/mL) by analyzing a set of standard solutions and they displayed good linear regression over the range (r = 0.9960–0.9999) (Table 2). The results of the instrumental precision investigation showed that the peak area RSD% of each component was less than or equal to 3.49% (Table 3), indicating that the precision of the instrument used in this study met the experimental requirements. Repeatability, intermediate precision, and their combined calculation showed the RSD% of each component was between 0.68% and 3.18% (Table 4 and Table 5), which proved that the method established above had better precision. The results of the stability investigation indicated that each compound was relatively stable within 12 h of storage in the sample chamber (RSD% ≤ 2.22%) (Table 6). The results of the average sample addition recovery of each component were 88.64–107.43%, RSD% at 2.82–6.22% (Table 7), which met the recovery limit requirements of the “Pharmacopoeia of the People’s Republic of China” (2020 edition) for simultaneous content determination of multiple components.

### 2.4. Simultaneous Quantitation of Fourteen Compounds in 15 Batches of GYJ Samples

The newly established method was used to calculate the content of the above-mentioned fourteen key compounds in 15 batches of GYJ samples by the method of accompanying mixed standard products. Two parallel samples were prepared for each batch, and each sample was acquired twice and accompanied by two needles of the standard. The content results were calculated by applying the ratio of the peak area to the ratio of the concentration and are shown in Table 8. Through the analysis of the standard deviation (SD) values of 15 batches of GYJ samples, the most significant difference is the Liquiritin and Glycyrrhizic acid, which were 0.83916 and 0.36383, respectively, indicating that different batches and different producing areas of Glycyrrhizae Radix et Rhizoma Praeparata Cum Melle have great quality differences. Next, the SD values of Rehmannioside D from Rehmanniae Radix Praeparata, Morroniside and Loganin from Corni Fructus, and Hyperoside from Cuscutae Semen were between 0.1 and 0.3, and the degree of dispersion was also large. The SD values of the remaining components were all less than 0.1, including Polygalaxanthone III, 3′,6-Disinapoylsucrose, and Tenuifolin from Polygalae Radix Praeparata, Ginsenoside Re, Ginsenoside Rg1, Ginsenoside Rb1 from Ginseng Radix et Rhizoma, Schisandrin from Schisandrae Chinensis Fructus, and Verbascoside from Rehmanniae Radix Praeparata. The results show that the quality of the different origins of the medicinal materials of Polygalae Radix Praeparata, Ginseng Radix et Rhizoma, and Schisandrae Chinensis Fructus was relatively stable. Therefore, in order to ensure the stability of preparation production during the later stage, the content range of the key ingredients contained in each medicinal material should be set as an exact and reasonable standard, and the quality control of the source should be strengthened.

### 2.5. Quality Evaluation by Cluster Analysis and Multivariate Statistical Analysis

The results of cluster heatmap analysis are shown in Figure 3A. The darker the color of the heat map indicates the higher the content of each component in the corresponding batch. The 15 batches of samples were divided into two categories, batches 1–5 as a group (S1–S5) and batches 6–15 (S6–S15) as a group. From the classification results, the differences in the content of the target chemical compounds were related to the quality variation of herbs from different origins. PCA was used to research the relationship or trend of similarity or differences among these samples, and the degree of clustering and dispersion of the samples could be observed from the score plot, as shown in Figure 3B; the results of PCA analysis were in accordance with the results of the clustering analysis. Due to the different origins of the formula’s herbs, there is a large difference in the content of some components, resulting in the poor stability of the GYJ samples, suggesting that control of the origins of the herbs or mixed batch inputting of herbs of different origins during the production of the preparation can be attempted. Further OPLS-DA analysis was performed to obtain the results of the VIP (Variable Importance for the Projection) ranking values of the components that have a high impact on the quality between different batches, as shown in Figure 3C,D, the top five compounds (VIP > 1) were Rehmannioside D, Loganin, Morroniside, Ginsenoside Re and 3′,6-Disinapoylsucrose. These five components originated from Radix Rehmanniae Praeparata, Cornus officinalis, Ginseng and Polygala Tenuifolia, which indicates that the quality stability of these four herbs has great influence on the stability of the GYJ preparation and are worth highlighting.

### 2.6. Overview of the Pharmacological Activities of Each Component

Owing to the multi-component and multi-target action characteristics of TCM, quality control research has always been a bottleneck on the road to modernization, limiting the benefit to all mankind. The selection of the index components in this study was, as far as possible, made to determine the active compounds related to the formula’s clinical efficacy. In the study exploring the compatibility mechanism of ShengDiHuang Decoction (SDHD) based on the in situ single-pass intestinal perfusion model; by analysing the effects of different concentrations, different pH, intestinal segments, protein inhibitors, and tight junction regulators on SDHD absorption, it was found that Rehmannioside D may undergo active transport, and may be a substrate of BCRP (breast cancer resistance protein) and MRP_2_ (multidrug resistance-associated protein 2) [14]. Recently, a study highlighted the protective role of Morroniside against H_2_O_2_-induced damage; further studies suggested that treatment with Morroniside decreased apoptosis, autophagy, and oxidative stress in rat ovarian granulosa cells through the PI3K (phosphatidylinositol 3-kinase)/AKT (serine/threonine-specific protein kinase)/mTOR (mechanistic target of rapamycin) pathway [15]. According to reports, Loganin mitigates Ang II-induced cardiac hypertrophy, at least partially, through inhibiting the JAK_2_ (Janus Kinase 2)_/_STAT_3_ (signal transducers and activators of transduction-3) and NF-κB (Nuclear factor-κB) signaling pathways and might be a novel effective agent for the treatment of cardiac hypertrophy and heart failure [16]. A quantibody array analysis demonstrated that Polygalaxanthone III downregulates inflammation in lipopolysaccharide-stimulated RAW264.7 macrophages [17]. Liquiritin reduces lipopolysaccharide-aroused HaCaT cell inflammation damage via the regulation of microRNA-31/MyD88 [18]. Hyperoside attenuates non-alcoholic fatty liver disease in rats via cholesterol metabolism and bile acid metabolism [19]. Ginsenoside Re ameliorates inflammation by inhibiting the binding of lipopolysaccharide to TLR4 (Toll-likereceptor4) on macrophages [20]. Research shows that the Ginsenoside Rg1 improved pathological damage in the ovary and uterus by increasing anti-oxidant and anti-inflammatory abilities whilst reducing the expression of senescence signaling pathways in POI (premature ovarian insufficiency) mouse models. Meanwhile, Glycyrrhizin possesses anti-inflammatory activity; hence, it is mostly used in traditional herbal medicine for the treatment and management of chronic diseases [21]. To sum up, quality control research on ingredients that may exert medicinal effects can better ensure the effectiveness and stability of the preparation. The simultaneous quality control of multiple components is more convincing than a single component, and it also reflects the theoretical system of TCM coordination and synergy.

## 3. Conclusions

In this study, an analytical method for the simultaneous quantitative analysis of fourteen components in GYJ samples based on the UPLC-MS/MS technique was developed for the first time. The method has the advantages of high sensitivity, high selectivity, and rapid analysis, which provides a reference for the quality control study and the Ancient Classical Formula research of GYJ granules’ development. In the meantime, through multivariate statistical analysis of the content determination results of 15 batches of GYJ samples in the three production areas, it was found that due to differences in the origins and batches of some medicinal materials, the dispersion degree of each batch was large. Therefore, in order to ensure the stability of subsequent preparation production, quality control research concerning the source of medicinal materials should be strengthened, such as the origin of the medicinal materials, collection period, traits, and specifications. Furthermore, the range standards for the upper and lower limits of content determination for key components should be fixed. Moreover, in the actual production process, manufacturers can also try to design a reasonably mixed batch inputting, so as to make better use of the medicinal materials and ensure the safety and effectiveness of the preparations.

## 4. Materials and Methods

### 4.1. Reagents and Materials

Chemical standards of Loganin, Polygalaxanthone III, Liquiritin, Hyperoside, Ginsenoside Re, Ginsenoside Rg1, Ginsenoside Rb1, Tenuifolin, Schisandrin were purchased from the National Institutes for Food and Drug Control (Beijing, China); Morroniside, Verbascoside, 3′,6-Disinapoylsucrose were offered by Chengdu Refensi Biotechnology Co., Ltd. (Chengdu, China); Rehmannioside D was obtained from Chengdu Purfield Biotechnology Co., Ltd. (Chengdu, China); Glycyrrhizic acid was acquired from Sichuan Vikki Biotechnology Co., Ltd. (Chengdu, China). The purity and batch number refer to Appendix A of the Appendix A. The purity of the above standards can be used for content determination. HPLC grade methanol and acetonitrile were provided by Fisher Scientific (Fair Lawn, NJ, USA), Chromatographic grade formic acid was acquired from Tianjin kemio Chemical Reagent Co., Ltd. (Tianjin, China). Water used throughout the experiments was prepared by Unique-R202 Multifunctional Ultrapure Water System (Xiamen, China). The other chemicals and solvents were all of analytical grade. The Chinese herbal medicines used in the preparation of the 15 batches of GYJ samples (S1–S15) were all sourced from authentic regions or main producing areas; the origin and batch number information is shown in Table 9. All medicine pieces are in compliance with the provisions of the “Pharmacopoeia of the People’s Republic of China” (2020 edition) section I of Chinese herbal medicine pieces, including properties, identification, inspection, content determination, processing, etc.

### 4.2. Preparation of Standard Solutions

A mixed standard stock solution (in 50% methanol) containing Rehmannioside D (1), Morroniside (2), Loganin (3), Polygalaxanthone III (4), Liquiritin (5), Hyperoside (6), Verbascoside (7), 3′,6-Disinapoylsucrose (8), Ginsenoside Re (9), Ginsenoside Rg1 (10), Ginsenoside Rb1 (11), Tenuifolin (12), Glycyrrhizic acid (13), Schisandrin (14) at concentrations of 20.410 µg/mL (1), 98.450 µg/mL (2), 52.975 µg/mL (3), 1.9650 µg/mL (4), 24.063 µg/mL (5), 29.888 µg/mL (6), 4.7850 µg/mL (7), 33.575 µg/mL (8), 5.3675 µg/mL (9), 2.9850 µg/mL (10), 6.7000 µg/mL (11), 3.5380 µg/mL (12), 37.300 µg/mL (13), and 8.9600 µg/mL (14) was prepared. The working standard solutions of different concentrations (0.4082–10.205 µg/mL (1); 1.9690–49.225 µg/mL (2); 1.0595–26.488 µg/mL (3); 0.0393–0.9825 µg/mL (4); 0.4813–12.031 µg/mL (5); 0.5978–14.944 µg/mL (6); 0.0957–2.3925 µg/mL (7); 0.6715–16.788 µg/mL (8); 0.1074–2.6838 µg/mL (9); 0.0597–1.4925 µg/mL (10); 0.1340–3.3500 µg/mL (11); 0.0708–1.7690 µg/mL (12); 0.7460–18.650 µg/mL (13); and 0.1792–4.4800 µg/mL (14)) were prepared by diluting the mixed standard solution with 50% methanol solution. All standard solutions were stored at 4 °C and filtered by a 0.22 µm membrane prior to injection.

### 4.3. Preparation of Sample Solutions

The daily dose of GYJ: 7.46 g of Ginseng Radix et Rhizoma, 14.92 g of Rehmanniae Radix Praeparata, 7.46 g of Dioscoreae Rhizoma, 5.60 g of Corni Fructus, 2.61 g of Polygalae Radix Praeparata (glycyrrhizae radix et rhizoma decoction processed), 5.60 g of Glycyrrhizae Radix et Rhizoma Praeparata Cum Melle, 2.00 g of Schisandrae Chinensis Fructus, 9.33 g of Cuscutae Semen (fried), was taken and placed in a Supor Decoction Casserole. A total of 400 mL of water was added, before soaking it for 60 min, then heating it using a Joyoung electric pottery stove, boiling on a strong fire for 10 min (1800 W), and then using a slow fire (400 W) for 75 min. The medicinal liquid was filtered while still warm (using a 120 mesh filter cloth) to obtain about 140 mL of decoction. Then, the decoction was frozen and rotated in a low-temperature absolute ethanol bath (−60 °C), so that the liquid evenly covered the inner wall of the freeze-dried bottle until it was completely solid. The freeze-dried bottle was stored at −80 °C for 24 h, and dried in a freeze dryer for 18 h to obtain the freeze-dried powder (yellow brown loose powder). Then, 0.4 g of freeze-dried powder was precisely weighed and placed in a 50 mL conical flask with a stopper. A total of 50 mL of 50% methanol solution was accurately added, the sample was then weighed and extracted ultrasonically for 45 min (250 W, 40 KHz). After ultrasonication, it was placed at room temperature, weighed again, and the lost weight was supplemented with 50% methanol solution. An appropriate amount of extract was then taken and passed through a 0.22 µm microporous membrane for UPLC-MS analysis.

### 4.4. Apparatus and Conditions

All samples were analyzed using UPLC (Waters Acquity^TM^ UPLC, Milford, MA, USA) with a Triple Quadrupole Mass Spectrometry System (Waters Synapt^TM^ TQD, Milford, MA, USA) (QQQ-MS). The separation was performed using the ACQUITY UPLC HSS T3 Column (2.1 × 100 mm, 1.8 µm) (Waters, Milford, MA, USA). The mobile phase was composed of acetonitrile + 0.1% formic acid (A) and water + 0.1% formic acid (B) at the flow rate of 0.4 mL/min. The column temperature was 35 °C, and the injection volume was 1 µL. The gradient elution of positive ion mode was as follows: 55–55% A at 0–3 min, 55–99% A at 3–5 min, and the re-equilibration time was 4 min; The gradient elution of negative ion mode was as follows: 5–15% A at 0–2 min, 15–15% A at 2–6 min, 15–50% A at 6–15 min, 50–99% A at 15–18 min, and the re-equilibration time was 5 min.

The ESI-MS spectra were acquired in the SIR mode under both positive and negative ion modes. The conditions for the ESI-MS analysis were set as follows: Capillary voltage: 3 kv/−3 kv; Cone voltage: 30 v/−30 v; Extractor voltage: 3 v/−3 v; Desolvation temperature: 350 °C; Desolvation gas flow: 650 L/h; Cone gas flow: 50 L/h; Source temperature: 150 °C; Data acquisition and processing were performed using Masslynx V4.2 software (Waters, USA). XSE105 Dual Range Analytical Balance (METTLER TOLEDO, Zurich, Switzerland); MODULYO freeze dryer (Thermo Fisher, Waltham, MA, USA); Pipettes (0.5 mL, 1 mL, 2 mL, 2.5 mL, 3 mL, 4 mL, 5 mL, BRAND Company, Wertheim, Germany); KQ-250DB Ultrasonic Cleaner (Kunshan Ultrasonic Instrument Co., Ltd., Kunshan, China); TB18A1 Supor Decoction Casserole (Zhejiang Supor Co., Ltd., Hangzhou, China); H22-X1 Joyoung electric ceramic stove (Joyoung Co., Ltd., Jinan, China).

### 4.5. Method Validation

Specificity, instrumental precision, linearity, repeatability, intermediate precision, stability, and accuracy were investigated during the method validation. Each negative control solution was prepared according to the method of preparation of the sample solutions for specificity investigation. After the instrument was acquired, they were compared and analyzed with the chromatogram of the standard and GYJ sample, which shows whether there were other chromatographic peaks and other interferences in the determination of the specific components in the sample using this method. The relative standard deviations (RSDs) were used to measure precision, stability, and repeatability. Instrument precision was calculated by collecting the mixed standard solution six times continuously and calculating the RSD% of the peak area of each component. For the calibration curves, six different concentrations of working standard solutions were analyzed in triplicate. The calibration curves were calculated by plotting the peak areas of each compound versus its concentration. To confirm the repeatability, six replicates of the same sample were extracted and analyzed. The operation of the intermediate precision experiment was the same as that of the repeatability, the operator and the operation date were changed. The intermediate precision experimental results and the repeatability results were combined to calculate the RSD%, indicating whether the precision of the method met the content determination requirements. For the stability test, the same sample was stored in the sample room and acquired by replicate analysis at 0, 2, 4, 6, 8, 10 and 12 h. The recovery test was performed to evaluate the accuracy of the method. One known amount (six samples in parallel) of standards was added into a certain number of samples and then these samples were extracted and analyzed using the established method. The recovery of each compound was calculated using the equation: Recovery = (Detected amount − Original amount)/Spiked amount × 100%.

### 4.6. Cluster Analysis and Multivariate Statistical Analysis

Cluster analysis is often used in the preliminary exploratory analysis of data, which can make data conclusions more concise and intuitive. Principal Component Analysis (PCA) is an algorithm for simplifying datasets and is often used to visualize similarities or differences in multivariate data, which is an unsupervised pattern recognition technique. Orthogonal Partial Least Squares-Discriminant Analysis (OPLS-DA) is a supervised mode, which reduces the dimensionality of the data and facilitates the screening of differential variables that contribute significantly to the grouping. The content results of fourteen key components in fifteen batches of GYJ samples (S1–S15) determined by the method established above were imported into Metware Cloud (One online data analysis platform, https://cloud.metware.cn/#/tools/tool-form?toolId=169 (accessed on 19 March 2022)) for advanced cluster analysis, advanced PCA, and OPLS-DA analysis.

## Figures and Tables

**Figure 1 molecules-27-08611-f001:**
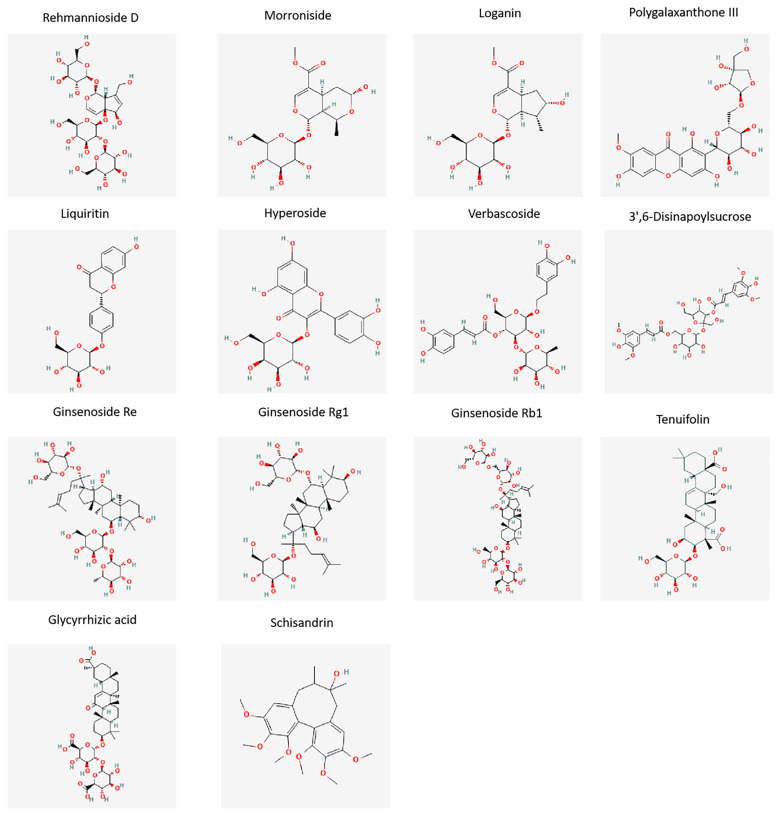
Structure information of the fourteen compounds analyzed in the GYJ samples.

**Figure 2 molecules-27-08611-f002:**
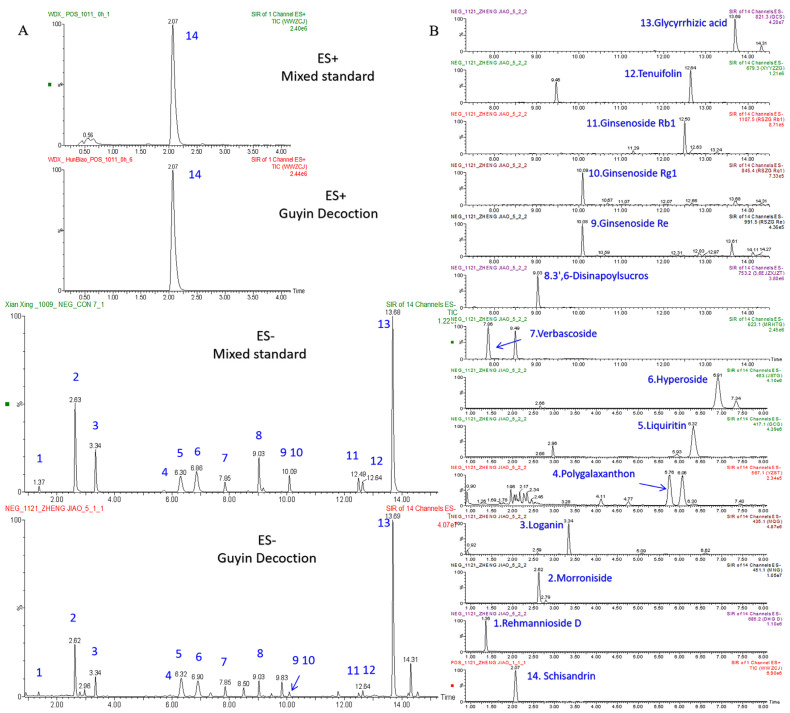
Representative chromatogram. (**A**) Total Ions Chromatograph (TIC) of the mixed standards compared with the GYJ samples. (**B**) Chromatograms of individual extracts of each compound in GYJ samples.

**Figure 3 molecules-27-08611-f003:**
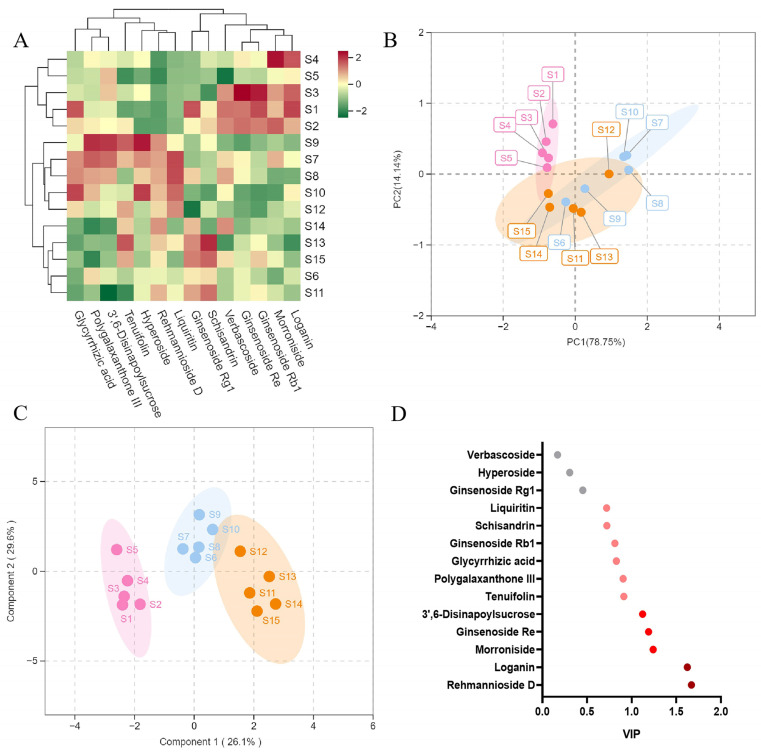
Results of cluster analysis and multivariate statistical analysis for the content determination of 14 analytes of fifteen batches (S1–S15) of GYJ samples. (**A**) Heat map of cluster analysis; (**B**) Results of PCA analysis; (**C**) Results of OPLS-DA analysis; (**D**) VIP Score Graph.

**Table 1 molecules-27-08611-t001:** Summary of UPLC-MS-SIR mode parameter information for 14 compounds.

SIR Mode	Compound Name	Mass (*m*/*z*)	Dwell (s)	Cone (V)
ES-	Rehmannioside D	685.2	0.018	55
ES-	Morroniside	451.1	0.018	30
ES-	Loganin	435.1	0.018	35
ES-	Polygalaxanthone III	567.1	0.018	75
ES-	Liquiritin	417.1	0.018	45
ES-	Hyperoside	463.0	0.018	60
ES-	Verbascoside	623.1	0.018	70
ES-	3′,6-Disinapoylsucrose	753.2	0.018	75
ES-	Ginsenoside Re	991.5	0.018	45
ES-	Ginsenoside Rg1	845.4	0.018	60
ES-	Ginsenoside Rb1	1107.5	0.018	90
ES-	Tenuifolin	679.3	0.018	65
ES-	Glycyrrhizic acid	821.3	0.018	90
ES+	Schisandrin	433.2	0.328	30

**Table 2 molecules-27-08611-t002:** Calibration curve, linear ranges, and r study results of 14 analytes.

Compound	Calibration Curve	Linear Range (µg/mL)	r
Rehmannioside D	y = 1856.2x + 162.47	0.4082–10.205	0.9999
Morroniside	y = 5663.3x + 9521	1.9690–49.225	0.9999
Loganin	y = 5837.6x + 3430.9	1.0595–26.488	0.9994
Polygalaxanthone III	y = 12,217x + 139.22	0.0393–0.9825	0.9999
Liquiritin	y = 11,344x + 3901.7	0.4813–12.031	0.9988
Hyperoside	y = 11,878x + 9390.3	0.5978–14.944	0.9960
Verbascoside	y = 19,206x + 1084.2	0.0957–2.3925	0.9993
3′,6-Disinapoylsucrose	y = 8154.2x + 2317.8	0.6715–16.788	0.9997
Ginsenoside Re	y = 12,765x + 325.7	0.1074–2.6838	0.9998
Ginsenoside Rg1	y = 16,044x + 175.01	0.0597–1.4925	0.9998
Ginsenoside Rb1	y = 15,886x + 203.89	0.1340–3.3500	0.9999
Tenuifolin	y = 24,931x + 1218.6	0.0708–1.7690	0.9988
Glycyrrhizic acid	y = 41,117x + 28,611	0.7460–18.650	0.9986
Schisandrin	y = 46,610x + 26,670	0.1792–4.4800	0.9966

**Table 3 molecules-27-08611-t003:** Instrument precision study results of 14 analytes.

Compound/Peak Area	1	2	3	4	5	6	RSD%
Rehmannioside D	2180	2162	2209	2230	2203	2046	3.03%
Morroniside	61,535	65,585	63,218	63,679	62,487	62,082	2.28%
Loganin	48,573	48,489	49,183	48,403	48,524	47,962	0.81%
Polygalaxanthone III	3501	3571	3557	3582	3457	3513	1.36%
Liquiritin	26,480	26,432	26,897	26,020	26,957	27,222	1.64%
Hyperoside	55,821	55,830	56,865	57,164	56,190	55,221	1.29%
Verbascoside	9523	9000	9509	9411	9947	9430	3.19%
3′,6-Disinapoylsucrose	43,871	43,924	46,404	44,412	41,994	45,135	3.31%
Ginsenoside Re	5886	6024	5879	5482	5802	5832	3.11%
Ginsenoside Rg1	4332	4297	4277	4328	4380	4356	0.87%
Ginsenoside Rb1	10,382	10,507	10,317	10,209	10,759	10,218	2.00%
Tenuifolin	9168	9394	9121	9258	9409	8543	3.49%
Glycyrrhizic acid	156,563	144,796	153,694	152,024	155,010	151,450	2.70%
Schisandrin	395,285	393,048	381,008	377,984	383,494	386,626	1.76%

**Table 4 molecules-27-08611-t004:** Repeatability study results of 14 analytes.

Compound	Sample-1 (mg/g)	Sample-2 (mg/g)	Sample-3 (mg/g)	Sample-4 (mg/g)	Sample-5 (mg/g)	Sample-6 (mg/g)	RSD%
Rehmannioside D	0.32113	0.32023	0.31742	0.31218	0.3173	0.32077	1.06%
Morroniside	2.91215	2.79783	2.8683	2.81242	2.85547	2.81521	1.52%
Loganin	1.53599	1.52903	1.52081	1.54322	1.52763	1.49539	1.08%
Polygalaxanthone III	0.05384	0.05257	0.05304	0.0518	0.0514	0.0534	1.79%
Liquiritin	0.51878	0.5036	0.51041	0.51195	0.50372	0.49634	1.55%
Hyperoside	0.71191	0.68372	0.69032	0.68591	0.67053	0.67489	2.12%
Verbascoside	0.14832	0.1476	0.14251	0.14216	0.14661	0.14786	1.90%
3′,6-Disinapoylsucrose	0.66255	0.64514	0.64891	0.64284	0.64037	0.63542	1.45%
Ginsenoside Re	0.07898	0.07593	0.07665	0.07623	0.07946	0.07656	1.96%
Ginsenoside Rg1	0.0863	0.08291	0.08512	0.08054	0.08308	0.08227	2.47%
Ginsenoside Rb1	0.08673	0.0868	0.08363	0.08353	0.08191	0.08369	2.32%
Tenuifolin	0.02412	0.02398	0.02403	0.02402	0.02421	0.02373	0.68%
Glycyrrhizic acid	1.29249	1.25455	1.25168	1.24285	1.26755	1.24067	1.53%
Schisandrin	0.22103	0.22152	0.21533	0.22478	0.22385	0.21874	1.57%

**Table 5 molecules-27-08611-t005:** Intermediate precision study results of 14 analytes.

Compound	Sample-1 (mg/g)	Sample-2 (mg/g)	Sample-3 (mg/g)	Sample-4 (mg/g)	Sample-5 (mg/g)	Sample-6 (mg/g)	RSD%	Combined Repeatability and Intermediate Precision (RSD%)
Rehmannioside D	0.31793	0.31807	0.31965	0.30476	0.29972	0.29848	3.18%	2.64%
Morroniside	2.84818	2.78914	2.81021	2.77910	2.81882	2.80366	0.86%	1.35%
Loganin	1.52682	1.53977	1.51064	1.52463	1.52649	1.49486	1.03%	1.02%
Polygalaxanthone III	0.05416	0.05272	0.05281	0.05269	0.05374	0.05289	1.17%	1.52%
Liquiritin	0.51802	0.51922	0.50072	0.50428	0.51141	0.51435	1.46%	1.49%
Hyperoside	0.69397	0.67495	0.67647	0.66746	0.66524	0.65313	2.03%	2.27%
Verbascoside	0.15130	0.14597	0.14822	0.14629	0.14771	0.14932	1.34%	1.76%
3′,6-Disinapoylsucrose	0.66602	0.66006	0.66075	0.65018	0.66738	0.66720	1.00%	1.74%
Ginsenoside Re	0.07692	0.07792	0.07820	0.07863	0.07875	0.07891	0.94%	1.58%
Ginsenoside Rg1	0.08542	0.08469	0.08397	0.08411	0.08526	0.08250	1.27%	1.96%
Ginsenoside Rb1	0.08749	0.08442	0.08543	0.08811	0.08529	0.08586	1.64%	2.18%
Tenuifolin	0.02405	0.02527	0.02446	0.02437	0.02381	0.02507	2.31%	1.95%
Glycyrrhizic acid	1.26006	1.28608	1.27006	1.25404	1.25920	1.25795	0.93%	1.24%
Schisandrin	0.22424	0.22423	0.23107	0.22125	0.22474	0.21432	2.45%	2.04%

**Table 6 molecules-27-08611-t006:** Stability study results of 14 analytes.

Compound	0 h (mg/g)	2 h (mg/g)	4 h (mg/g)	6 h (mg/g)	8 h (mg/g)	10 h (mg/g)	12 h (mg/g)	RSD%
Rehmannioside D	0.28387	0.29758	0.28751	0.27986	0.28424	0.28517	0.28306	1.97%
Morroniside	2.73107	2.61553	2.63035	2.61832	2.67367	2.66227	2.64084	1.53%
Loganin	1.42990	1.43415	1.45469	1.43699	1.42918	1.40613	1.43684	1.01%
Polygalaxanthone III	0.04613	0.04684	0.04714	0.04891	0.04796	0.04685	0.04878	2.22%
Liquiritin	0.47113	0.46881	0.47472	0.46043	0.47498	0.46801	0.47534	1.14%
Hyperoside	0.58354	0.57778	0.56404	0.55416	0.56335	0.55998	0.56784	1.80%
Verbascoside	0.12900	0.13135	0.12806	0.12895	0.12647	0.13206	0.13151	1.60%
3′,6-Disinapoylsucrose	0.57402	0.56734	0.58351	0.59466	0.59594	0.58482	0.56461	2.15%
Ginsenoside Re	0.07292	0.07137	0.07278	0.07188	0.06977	0.0722	0.06956	1.90%
Ginsenoside Rg1	0.07888	0.07499	0.07763	0.07661	0.07514	0.07685	0.07432	2.12%
Ginsenoside Rb1	0.07890	0.07639	0.07756	0.07889	0.0775	0.07682	0.07864	1.30%
Tenuifolin	0.02164	0.02209	0.02203	0.02193	0.02174	0.02252	0.02193	1.29%
Glycyrrhizic acid	1.15635	1.15132	1.1506	1.14429	1.15882	1.15643	1.13918	0.62%
Schisandrin	0.21346	0.21495	0.21659	0.21666	0.21509	0.21144	0.21711	0.94%

**Table 7 molecules-27-08611-t007:** Recovery results of 14 analytes.

Compound	Number	Original Amount (mg)	Spiked Amount (mg)	Detected Amount (mg)	Recovery (%)	Mean	RSD/% (*n* = 6)
Rehmannioside D	1	0.04407	0.05016	0.09262	96.78%	93.49%	2.82%
	2	0.04490		0.09072	91.34%		
	3	0.04493		0.09023	90.30%		
	4	0.04404		0.09129	94.22%		
	5	0.04493		0.09315	96.13%		
	6	0.04507		0.09131	92.19%		
Morroniside	1	0.62622	0.54037	1.17592	101.73%	96.96%	4.52%
	2	0.63802		1.17246	98.90%		
	3	0.63843		1.14833	94.36%		
	4	0.62568		1.17455	101.57%		
	5	0.63840		1.13166	91.28%		
	6	0.64038		1.14775	93.89%		
Loganin	1	0.31094	0.31531	0.65369	108.71%	100.99%	4.78%
	2	0.31679		0.63562	101.12%		
	3	0.31700		0.63492	100.83%		
	4	0.31067		0.63647	103.33%		
	5	0.31698		0.62244	96.87%		
	6	0.31796		0.61783	95.10%		
Polygalaxanthone III	1	0.00828	0.01138	0.02072	109.37%	101.46%	6.22%
	2	0.00844		0.01963	98.38%		
	3	0.00844		0.01944	96.67%		
	4	0.00827		0.02071	109.30%		
	5	0.00844		0.01981	99.91%		
	6	0.00847		0.01929	95.13%		
Liquiritin	1	0.09114	0.10973	0.19988	99.10%	101.09%	4.30%
	2	0.09286		0.20724	104.25%		
	3	0.09292		0.20754	104.46%		
	4	0.09106		0.20721	105.86%		
	5	0.09291		0.19734	95.17%		
	6	0.09320		0.20041	97.71%		
Hyperoside	1	0.15628	0.18152	0.31016	84.77%	88.64%	3.87%
	2	0.15923		0.32162	89.46%		
	3	0.15933		0.32849	93.19%		
	4	0.15615		0.32266	91.73%		
	5	0.15932		0.31400	85.21%		
	6	0.15982		0.31861	87.48%		
Verbascoside	1	0.02123	0.02553	0.04767	103.57%	100.48%	3.29%
	2	0.02163		0.04829	104.45%		
	3	0.02164		0.04648	97.31%		
	4	0.02121		0.04596	96.96%		
	5	0.02164		0.04676	98.41%		
	6	0.02171		0.04779	102.16%		
3′,6-Disinapoylsucrose	1	0.10012	0.11174	0.21203	100.15%	100.42%	3.27%
	2	0.10200		0.21945	105.10%		
	3	0.10207		0.21586	101.83%		
	4	0.10003		0.21002	98.43%		
	5	0.10206		0.20871	95.44%		
	6	0.10238		0.21584	101.53%		
Ginsenoside Re	1	0.00994	0.01203	0.02259	105.12%	104.78%	4.49%
	2	0.01013		0.02383	113.89%		
	3	0.01013		0.02236	101.64%		
	4	0.00993		0.02210	101.17%		
	5	0.01013		0.02266	104.13%		
	6	0.01017		0.02253	102.75%		
Ginsenoside Rg1	1	0.01203	0.01103	0.02369	105.70%	107.43%	4.06%
	2	0.01226		0.02481	113.77%		
	3	0.01227		0.02449	110.75%		
	4	0.01202		0.02335	102.66%		
	5	0.01227		0.02423	108.43%		
	6	0.01231		0.02370	103.27%		
Ginsenoside Rb1	1	0.01126	0.01222	0.02396	103.93%	103.97%	4.11%
	2	0.01147		0.02508	111.35%		
	3	0.01148		0.02415	103.72%		
	4	0.01125		0.02322	97.96%		
	5	0.01147		0.02413	103.56%		
	6	0.01151		0.02414	103.31%		
Tenuifolin	1	0.00462	0.00519	0.00999	103.48%	99.15%	3.97%
	2	0.00471		0.00975	97.23%		
	3	0.00471		0.00966	95.51%		
	4	0.00462		0.00981	100.21%		
	5	0.00471		0.00962	94.73%		
	6	0.00472		0.01010	103.73%		
Glycyrrhizic acid	1	0.21756	0.24525	0.44908	94.40%	92.34%	2.87%
	2	0.22166		0.45797	96.36%		
	3	0.22181		0.44341	90.36%		
	4	0.21738		0.43690	89.51%		
	5	0.22179		0.44419	90.68%		
	6	0.22248		0.44992	92.74%		
Schisandrin	1	0.04957	0.04476	0.09123	93.09%	91.22%	2.97%
	2	0.05050		0.09265	94.17%		
	3	0.05053		0.09110	90.64%		
	4	0.04952		0.08992	90.26%		
	5	0.05053		0.08928	86.58%		
	6	0.05069		0.09211	92.55%		

**Table 8 molecules-27-08611-t008:** Content determination results of 14 analytes in fifteen batches of GYJ samples.

Content (mg/g, *n* = 2)																
	S1	S2	S3	S4	S5	S6	S7	S8	S9	S10	S11	S12	S13	S14	S15	SD
Rehmannioside D	0.26021	0.26517	0.30544	0.23689	0.18487	0.58298	0.83644	0.81856	0.87372	0.77825	0.79725	0.71777	0.74770	0.86893	0.67400	0.27359
Morroniside	2.62522	2.82448	2.69635	2.99392	2.46303	2.18398	2.64711	2.18389	2.30279	2.12395	2.38980	2.32503	2.26280	2.34720	2.44680	0.25363
Loganin	1.55306	1.43519	1.52310	1.54024	1.36209	1.29946	1.34021	1.19817	1.21662	1.36191	1.23902	1.31137	1.13805	1.24123	1.22655	0.13082
Polygalaxanthone III	0.04645	0.05232	0.04380	0.05116	0.04430	0.05436	0.06872	0.06084	0.07404	0.05811	0.03849	0.04437	0.03731	0.02709	0.02692	0.01355
Liquiritin	0.55860	0.47562	0.56533	0.46908	0.53224	0.99557	2.63008	2.60939	1.39165	2.40877	1.22428	2.15027	1.44371	0.48911	0.48426	0.83916
Hyperoside	0.65880	0.63195	0.81321	0.81044	0.65848	0.78963	0.94112	0.86762	1.06844	1.02875	0.79034	0.74206	0.73911	0.72920	0.71311	0.13016
Verbascoside	0.16081	0.15508	0.15121	0.10594	0.05612	0.09847	0.13244	0.14653	0.08705	0.12370	0.09823	0.09399	0.07120	0.15331	0.13059	0.03299
3′,6-Disinapoylsucrose	0.49389	0.51598	0.56075	0.50473	0.56824	0.49123	0.63219	0.58257	0.67184	0.49296	0.31598	0.50248	0.42314	0.38143	0.43481	0.09272
Ginsenoside Re	0.06604	0.06684	0.07950	0.05246	0.04317	0.04820	0.05148	0.04944	0.03211	0.03327	0.04824	0.03420	0.04461	0.03665	0.04658	0.01342
Ginsenoside Rg1	0.09848	0.07674	0.06300	0.06137	0.06128	0.07863	0.06219	0.08167	0.06305	0.05879	0.08731	0.04565	0.09070	0.08427	0.09229	0.01540
Ginsenoside Rb1	0.09314	0.08419	0.10099	0.06016	0.04954	0.04994	0.05074	0.05169	0.04190	0.03186	0.06376	0.03319	0.06085	0.04985	0.07120	0.02037
Tenuifolin	0.02034	0.02494	0.02309	0.02345	0.01945	0.02239	0.02627	0.02163	0.02878	0.02342	0.01925	0.02687	0.02902	0.02584	0.02678	0.00316
Glycyrrhizic acid	1.91695	1.52411	1.27045	1.17990	1.20443	0.97810	1.74063	1.75546	1.34449	1.96704	0.92250	1.51239	0.91537	0.98217	1.07492	0.36383
Schisandrin	0.21841	0.22412	0.17753	0.20762	0.19557	0.22222	0.19976	0.20067	0.21398	0.18160	0.24425	0.20641	0.25835	0.18588	0.24667	0.02402

**Table 9 molecules-27-08611-t009:** Origin and batch numbers of the Chinese herbal medicines in the fifteen batches of GYJ samples (S1–S15).

**Chinese Herbal Medicine**	**Ginseng Radix et Rhizoma**		**Rehmanniae Radix Praeparata**		**Dioscoreae Rhizoma**		**Corni Fructus**	
**Sample** **Number**	**Origin**	**Batch Number**	**Origin**	**Batch Number**	**Origin**	**Batch Number**	**Origin**	**Batch Number**
S1	Jingyu, Jilin Province	17112311	Xiangfen, Shanxi Province	17102711	Anguo, Hebei Province	18011611	Nanyang, Henan Province	1709061
S2		17112312		17102712		18011612		1709062
S3		17112313		17102713		18011613		1709063
S4		17112314		17102714		18011614		1709064
S5		17112315		17102715		18011615		1709065
S6	Xinbin, Liaoning Province	201001	Jiaozuo, Henan Province	20191201	Anyang, Henan Province	20201101	Danfeng, Shaanxi Province	180311
S7		201002		20191202		20201102		180312
S8		201003		20191203		20201103		180313
S9		201004		20191204		20201104		180314
S10		201005		20191205		20201105		180315
S11	Tonghua, Jilin Province	201006	Wenxi, Shanxi Province	20191201	Li xian, Hebei Province	20201101	Luoyang, Henan Province	180321
S12		201008		20191202		20201102		180322
S13		201009		20191203		20201103		180323
S14		201010		20191204		20201104		180324
S15		201101		20191205		20201105		180325
**Chinese Herbal Medicine**	**Polygalae Radix**		**Glycyrrhizae Radix et Rhizoma Praeparata Cum Melle**		**Schisandrae Chinensis Fructus**		**Cuscutae Semen**	
**Sample** **Number**	**Origin**	**Batch Number**	**Origin**	**Batch Number**	**Origin**	**Batch Number**	**Origin**	**Batch Number**
S1	Xintai, Shandong Province	18010311	Longxi, Gansu Province	17111411	Ji’an, Jilin Province	1710271	Pingluo, Ningxia Province	17120911
S2		18010312		17111412		1710272		17120912
S3		18010313		17111413		1710273		17120913
S4		18010314		17111414		1710274		17120914
S5		18010315		17111415		1710275		17120915
S6	Yulin, Shaanxi Province	20191001	Tongxin, Ningxia	TX201101	Qinglong, Hebei Province	180311	Anguo, Hebei Province	20201101
S7		20191002		TX201102		180312		20201102
S8		20191003		TX201103		180313		20201103
S9		20191004		TX201104		180314		20201104
S10		20191005		TX201105		180315		20201105
S11	Wenxi, Shanxi Province	190372001	Hangjinqi, Inner Mongolia	201101	Jingyu, Jilin Province	180321	Pingluo, Ningxia Province	20201101
S12		190373001		201102		180322		20201102
S13		190374001		201103		180323		20201103
S14		190375001		201902		180324		20201104
S15		190376001		201904		180325		20201105

## Data Availability

All data can be found in this paper.

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
