# Peer review of "Development and Validation of a UPLC-MS/MS Method for the Quantification of Components in the Ancient Classical Chinese Medicine Formula of Guyinjian"

_molecules, 2022, doi:10.3390/molecules27238611_

Round 1

Reviewer 1 Report

1-Figure 1 would be better removed.

2-Why is the Material Method section before the conclusion and discussion section? It should be corrected.

3-It would be better if it was called UPLC-MS/MS Method instead of UPLC-QQQ-MS Method in the title.

4-Ethical approval number and information should be written.

5-English should be revised.

6-There are some grammatical errors, they should be corrected.

Round 2

Reviewer 1 Report

1-Ethical approval number has not been written or ethical approval has not been obtained, it should be obtained.

2-The material method section should be placed after the introduction, not done.

Reviewer 2 Report

All the remarks raised are taken into consideration so I give a favorable opinion for the publication of this interesting work.